# Frequency-specific cortico-subcortical interaction in continuous speaking and listening

**Omid Abbasi[1]\*, Nadine Steingräber[1], Nikos Chalas[1,2], Daniel S Kluger[2], Joachim Gross[2]**

[1]Institute for Biomagnetism and Biosignal Analysis, University of Münster, Münster, Germany; [2]Otto-Creutzfeldt-Center for Cognitive and Behavioral Neuroscience, University of Münster, Münster, Germany

## eLife assessment

Abbasi and colleagues use Granger causality to explore the cortico-subcortical dynamics during speaking and listening. They find **valuable** evidence for bi-directional connectivity in distinct frequency bands as a function of behaviour, but currently offer **incomplete** support for the validity of their analyses and the predictive coding interpretation of their results.

**\*For correspondence:**
abbasi@wwu.de

**Competing interest:** The authors declare that no competing interests exist.

## Abstract

Speech production and perception involve complex neural dynamics in the human brain. Using magnetoencephalography, our study explores the interaction between cortico-cortical and cortico-subcortical connectivities during these processes. Our connectivity findings during speaking revealed a significant connection from the right cerebellum to the left temporal areas in low frequencies, which displayed an opposite trend in high frequencies. Notably, high-frequency connectivity was absent during the listening condition. These findings underscore the vital roles of cortico-cortical and cortico-subcortical connections within the speech production and perception network. The results of our new study enhance our understanding of the complex dynamics of brain connectivity during speech processes, emphasizing the distinct frequency-based interactions between various brain regions.

## Introduction

Human communication can be described as two dynamic systems (i.e., speaker and listener) that are coupled via sensory information (*Silbert et al., 2014*) and operate according to principles of active inference by minimizing prediction errors (*Friston and Frith, 2015*). Within this framework, the speaker's predictive processing model steers speech production, allowing adjustments in volume, speed, or articulation based on proprioceptive and auditory feedback. Similarly, the listener's predictive processing model generates anticipations about upcoming speech which are continually updated and compared with incoming sensory data. The implementation of this model in the human brain was shown to be associated with brain rhythms (*Abbasi et al., 2023*; *Arnal and Giraud, 2012*; *Park et al., 2015*).

Brain rhythms have been extensively investigated during continuous speech perception. A consistent finding in these studies is the synchronization of frequency-specific brain activity, with the rhythmic amplitude modulation in continuous speech (*Giraud and Poeppel, 2012*; *Gross et al., 2013b*). The exploration of the brain networks that underpin speech perception has also revealed a stronger causal influence from higher-order brain regions (such as the left inferior frontal gyrus and left motor areas)

to the auditory cortex for intelligible speech compared to unintelligible speech (*Park et al., 2015*). In the domain of speech production, both invasive (*Llorens et al., 2011*; *Ozker et al., 2022*; *Riès et al., 2017*) and non-invasive (*Ganushchak et al., 2011*; *Janssen et al., 2020*) electrophysiological studies have made significant contributions due to their superior temporal resolution compared to fMRI. However, it is important to note that invasive recordings offer only limited spatial coverage from specific recording sites in patients. Magnetoencephalography (MEG) has been used by several research teams to explore the intricate relationship between brain frequency-specific dynamics and speech production. Ruspantini et al. observed notable coherence between MEG activity originating from the motor cortex and Electromyography (EMG) activity in lip muscles, with a peak frequency of approximately 2–3 Hz (*Ruspantini et al., 2012*). In a separate investigation conducted by Alexandrou et al., they identified high-frequency band modulation (60–90 Hz) within bilateral temporal, frontal, and parietal brain regions (*Alexandrou et al., 2017*). In our recent study, we also explored speech production and perception using MEG to map the cortical networks engaged during these two experimental conditions (*Abbasi et al., 2023*). We reported significant connectivity from the motor area to superior temporal gyrus (STG) in lower-frequency bands (up to beta) during speaking, and the reverse pattern in high gamma frequencies (*Abbasi et al., 2023*).

Subcortical areas also contribute in updating the brain's internal model during speech production and perception. Thalamic nuclei serve as central nodes for circuits associated with language processing, interacting with the frontal cortex, basal ganglia, cerebellum, and dopaminergic groups. Notably, the motor-related thalamic nuclei link the basal ganglia and cerebellum with the frontal cortex, contributing to both motor and cognitive aspects of language (*Barbas et al., 2013*; *Silveri, 2021*). Relatedly, several previous studies have reported the cerebellum's role in several cognitive functions such as speech production and perception (*Giraud et al., 2007*; *Skipper and Lametti, 2021*) movement coordination, timing, motor programming, speech motor control, and sensory prediction (*Parrell et al., 2017*). Lesion studies, neuroimaging investigations, and brain stimulation studies have consistently linked the cerebellum to critical aspects of timing and predictive processing in speech perception (*Ackermann, 2008*; *Manto et al., 2012*; *Parrell et al., 2017*). Temporo-cerebellar coupling has been suggested to support the continuous updating of internal models of the auditory environment (*Kotz et al., 2014*). In a more recent study, Stockert et al. reported bidirectional temporo-cerebellar anatomical connectivity (*Stockert et al., 2021*). The authors suggested that these anatomical temporo-cerebellar connections may support the encoding and modeling of rapidly modulated auditory spectro-temporal patterns.

However, to the best of our knowledge no study has so far investigated spectrally resolved, directed functional coupling between cortical and subcortical brain areas during listening and speaking. In this MEG study, we intend to investigate the specific frequency-related connections between both cortical and subcortical brain regions while participants engaged in speaking and listening tasks. Participants answered seven questions, each lasting 60 s (speaking condition), and also listened to recordings of their own speech from the previous speaking session (listening condition; see Methods for details). Initially, we identified the cortical and subcortical brain regions involved in speech perception and production through meta-analysis studies. Subsequently, we examined the brain network and directed connections within it to shed light on the cognitive processes underlying listening and speaking. This involved a direct examination of frequency-specific communication channels for both feedforward and feedback signaling during speech production and perception using multivariate Granger causality. Our connectivity findings confirmed the participation of subcortical regions such as the thalamus and cerebellum in both speech production and perception. Our analysis using directed Granger causality revealed stronger connectivity from the cerebellum to the auditory area at slower frequencies (below 40 Hz). This connectivity pattern might reflect the influence of cerebellar predictions on auditory processing. Conversely, connectivity in the opposite direction, from the auditory area to the cerebellum, was stronger at faster frequencies (above 40 Hz). This pattern could potentially be indicative of prediction error signals being relayed back to the cerebellum for updating its predictions.

## Results

In this study, we investigated the connectivity between brain regions that are involved in speech production and perception. OurROI-based (Region of Interest) analysis utilized the automatic meta-analysis provided by neurosynth.org using the term 'speech' that resulted in 642 fMRI studies

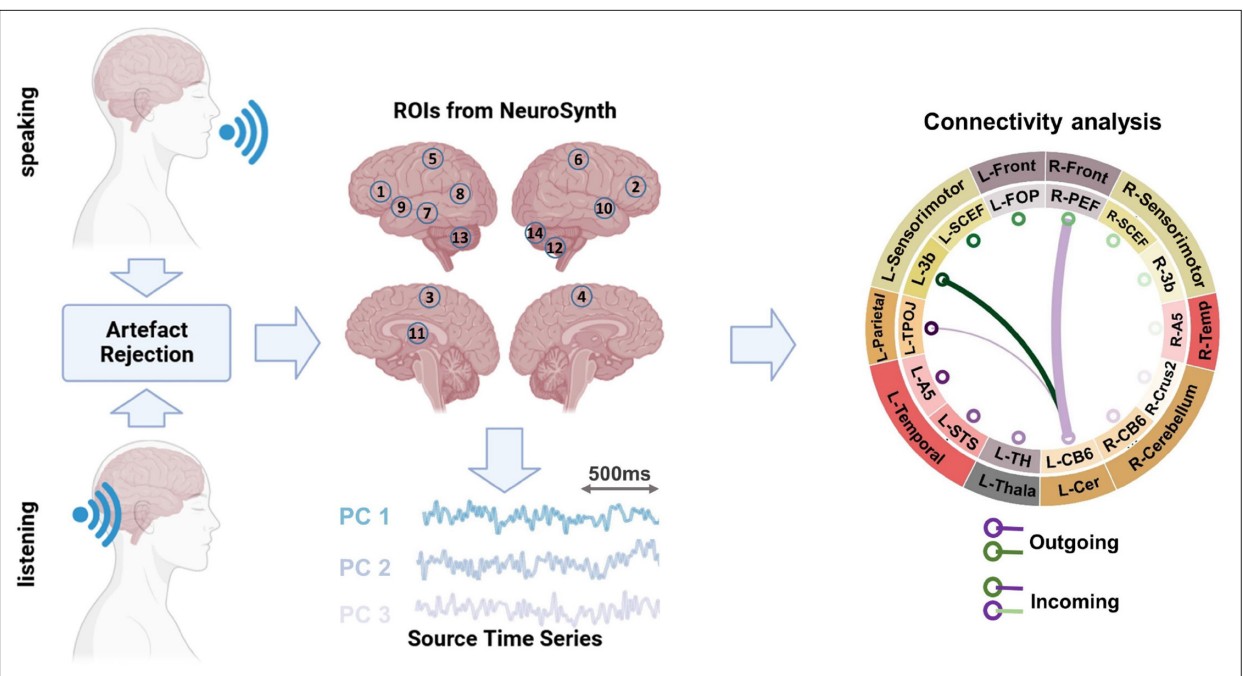

**Figure 1.** Methodological pipeline. Thirty participants answered seven given questions (60 s each; *speaking* condition) as well as listened to audio-recordings of their own voice from previous sessions (*listening* condition) while magnetoencephalography (MEG) data were recorded. Artifacts were removed from the recorded MEG data (*Abbasi et al., 2021*). Individual MRIs were used to estimate source models per participant which were interpolated to a template volumetric grid. Relevant areas in speech production and perception networks were identified from Neurosynth.org platform. Fourteen corresponding anatomical parcels in HCP and AAL atlases were identified: L-FOP (1), R-PEF+6v (2), L-SCEF (3), R-SCEF (4), L-3b (5), R-3b (6), L-STS (7), L-TPOJ1 (8), L-A5 (9), R-A5 (10), L-TH (11), R-CB-Crus2 (12), L-CB-6 (13), and R-CB-6 (14). For each identified parcel, estimated source time-series were extracted. Next, using a blockwise approach, we considered the first three singular value decomposition (SVD) components of each parcel as a block and estimated the connectivity between each pair of parcels using a multivariate nonparametric Granger causality approach (mGC; *Schaum et al., 2021*). In this study, the connectivity results are presented using connectogram plots. In the connectograms, nodes represent the brain areas and edges represent the strength and direction of the connections between them. The thickness of the edges indicates the magnitude of the *t*-values, while color indicates the directionality of the connectivity. In other words, when node A connects to node B, the edge will have the same color as node A, and vice versa when node B connects to node A. Note that only significant connections are shown in the connectograms (p < 0.05, cluster correction). For instance in the illustrated connectogram, the purple edge between L-CB6 and R-PEF shows significant connectivity from L-CB6 to R-PEF. R = right, L = left.

highlighting the brain areas involved in speech production and perception. We extracted the MNI coordinates of active areas from the resulting neurosynth statistical map, resulting in 14 parcels. For each voxel in a given parcel, the beamformer-derived time-series for all the three source orientations were subjected to singular value decomposition (SVD). In every parcel, we used the three strongest components for further analysis. For estimating pairwise functional connectivity between all parcels (*n* = 14), we used multivariate nonparametric Granger causality (mGC; *Schaum et al., 2021*) and related them to the hallmarks of predictive processing models during continuous speaking and listening (see *Figure 1*). The method uses three-dimensional representations of parcel activity and thus estimates connectivity more accurately than traditional one-dimensional representations (*Chalas et al., 2022*).

The calculation of mGC between two parcels yielded two spectra reflecting both directions (A->B and B->A). Next, we computed the directed asymmetry index (DAI) for each pair of spectra (*Bastos et al., 2015*). This measure captures the predominant direction of mGC between two parcels based on the relative difference between both directions (see 'Methods' for more information). A positive DAI indicates the dominant directionality from A toward B, while a negative DAI indicates the opposite directionality. First, we used group statistics to identify brain areas where DAI values in specific frequency bands in the speaking as well as listening conditions differed significantly from zero. We performed group statistics from 0 to 100 Hz. Next, we defined the following canonical frequency bands: Delta/Theta (0–7 Hz), alpha (7–13 Hz), beta (15–30 Hz), gamma (30–60 Hz), and high gamma (60–90 Hz).

Using connectogram plots, we visualized the connectivity between the selected 14 ROIs and examined neural mechanisms associated with speech production and perception, as well as predictive processing models. The number of outgoing and incoming edges for each node was also calculated using the brain connectivity toolbox and represented as the node strength (*Rubinov and Sporns, 2010*). The upper row of *Figure 2* illustrates the strength for three different nodes in the speaking condition. In L-SCEF (pre-supplementary motor area) and R-CB6 (right cerebellum lobule VI), the number of outgoing edges in low frequencies is higher compared to incoming edges. However, we observed a reverse pattern in high frequencies. Interestingly, in the left STG (L-A5), there are more incoming connections compared to outgoings in low frequencies, and this changes in the higher frequencies, with more outgoing connections seen in this area. This result indicates that in low frequencies, the sensorimotor area and cerebellum predominantly transmit information, while in higher frequencies, they are more involved in receiving it. On the other hand, the left STG receives information in low frequencies and transmits it in high frequencies. The connectograms in the middle parts of *Figure 2* illustrate the connections of the left sensorimotor area (left plots; L-3b and L-SCEF) and the right cerebellum (right plots) to other cortical and subcortical parcels in low (top: 7–20 Hz) and high (bottom: 60–90 Hz) frequencies during the speaking condition. In low frequencies, both the sensorimotor area and the right cerebellum primarily send information to other parcels, including the left and right temporal regions, while in the high frequencies, these nodes primarily receive information. Interestingly, our connectivity results illustrate that the top–down effects from higher-order cortical areas to lower-order areas during speaking occur in distinct frequency bands. *Figure 2* (lower panels) depicts that the strongest top–down connectivity from the left pre-supplementary motor area (pre-SMA; L-SCEF) to lower-order cortical areas such as L-A5 occur in the theta band (peak at 5–7 Hz). However, the top–down connectivity from L-3b to L-A5 was strongest in two distinct theta (peak at 7 Hz) and beta (peak at 21 Hz) frequency bands. Moreover, these results revealed significant bottom–up connectivity from lower-order cortical areas, namely L-A5 to left sensorimotor area (L-3b and L-SCEF) in high-frequency bands. As we previously showed (*Abbasi et al., 2023*), this striking reversal indicates a dissociation of bottom–up and top–down information flow during speaking in distinct frequency bands. Signals communicated top–down are predominantly transmitted in low-frequency ranges, while those communicated bottom–up are transmitted in high-frequency ranges. Notably, *Figure 2* (bottom-right panel) also depicts directional connectivity from the right cerebellum lobule VI (R-CB6) to the left STG (L-A5) in frequencies below 50 Hz. In contrast, in frequencies above 50 Hz, we observed reverse signaling from the left STG to the right cerebellum during speaking.

The connectograms in *Figure 3* provide a comprehensive overview of significant connectivities across various frequency bands among the selected fourteen brain areas during speaking and listening conditions. Specifically highlighting the first column's connectograms during speaking, we observe comparable connectivity patterns to those of the left sensorimotor area. Connections from the right higher-order cortical regions, such as R-SCEF and R-3b, toward lower-order areas like R-A5, are noticeable in low-frequency bands (<30 Hz). Conversely, for higher frequencies (>30 Hz), we noticed the reverse directional connections, from lower- to higher-order brain regions. Furthermore, *Figure 4 (upper panel)* reveals strong top–down connectivity from bilateral 3b and left FOP (frontal opercular area) to bilateral A5 and left STS (superior temporal sulcus), particularly prominent in two distinct frequency bands: theta (peak at 7 Hz) and beta (peak at 21 Hz). Additionally, similar to the left hemisphere, our results in *Figure 4 (upper panel)* show significant bottom–up connectivity from lower-order cortical areas (bilateral A5 and left STS) to higher-order cortical regions (bilateral 3b and bilateral SCEF) across both low- and high-frequency bands.

We observed a similar pattern during the listening condition compared to the speaking condition, including reversed directionality between low and high frequencies (*Figure 3*, second column). However, both top–down effects in the low-frequency band and bottom–up effects in the high-frequency band were less pronounced during listening compared to speaking. The third column in *Figure 3* as well as the third panel of *Figure 4* confirm these results by showing the direct comparisons of DAI connectivity between speaking and listening. This figure illustrates significant differences in connectivity between speaking and listening conditions in different frequency bands. Specifically, stronger top–down signals originating from higher-order cortical areas (such as bilateral SCEF and 3b) toward lower-order brain areas, such as bilateral A5 and left STS were observed during speaking compared to listening. Conversely, the individual spectrally resolved DAI plots for the listening

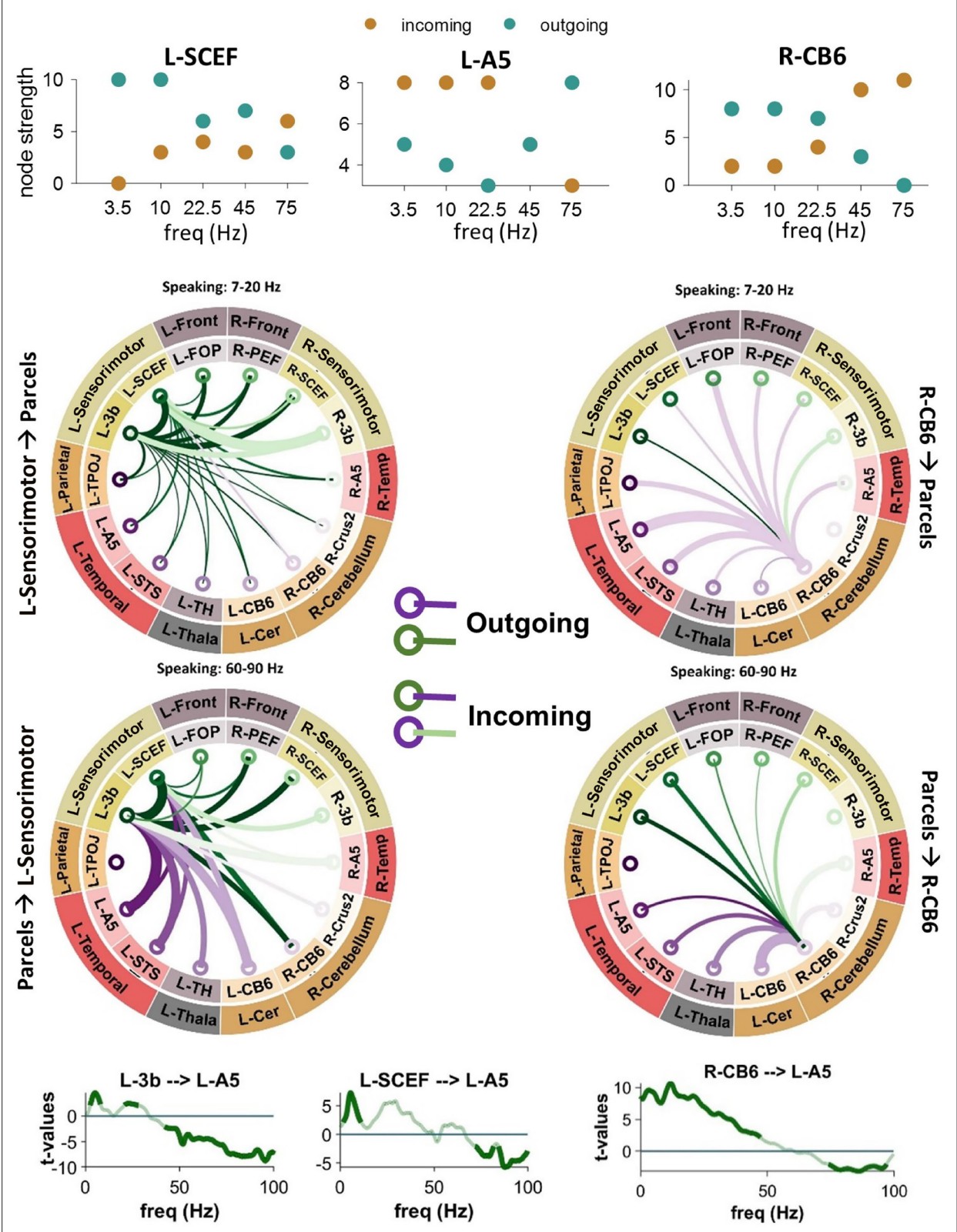

**Figure 2.** Connectivity analysis between the sensorimotor, cerebellum, and superior temporal gyrus (STG). Upper panels illustrate the strength for L-SCEF, L-A5, and R-CB6 nodes in the speaking condition. In the middle part, connectograms illustrate connections between the left sensorimotor area (left plots) and the right cerebellum (right plots) during speaking to other cortical and subcortical parcels at low frequencies (top: 7–20 Hz) and high frequencies (bottom: 60–90 Hz). Lower panels illustrate spectrally resolved directed asymmetry index (DAI) from L-3b (left spectra), L-SCEF (middle

*Figure 2 continued on next page*

*Figure 2 continued*

spectra), and R-CB6 (right spectra) to L-A5. The results represented in the whole figure are significant connectivity patterns that passed a cluster-based permutation test (p <.05, cluster correction).

conditions, along with the connectograms, indicate the presence of significant connectivity solely from low-order areas (such as bilateral A5) toward higher-order cortical areas (bilateral SCEF and 3b) during the speaking condition, as opposed to the listening condition. More detailed spectrally resolved DAI between all ROIs and both conditions are illustrated in *Figure 4*.

Notably, we found significant connections between subcortical and cortical areas, indicating their involvement in speech production and perception networks. *Figure 4* depicts the connectivity from the right cerebellum to other cortical and subcortical parcels. In contrast with what we reported for the speaking condition, during listening, there is only a significant connectivity in low frequency to the left temporal area but not a reverse connection in the high frequencies. However, a direct comparison between speaking and listening conditions did not reveal a significant difference. Our analysis also revealed robust connectivity between the right cerebellum and the left parietal cortex, evident in both speaking and listening conditions, with stronger connectivity observed during speaking. Notably, *Figure 4* depicts information flows from the cerebellum to the parietal areas in low-frequency ranges. There is also an intriguing connection between the left thalamus and right cerebellum, which reveals a distinct pattern of connectivity. Specifically, we found significant connectivity from the cerebellum to the thalamus in the low-frequency range during speaking, while the opposite pattern was observed in the high-frequency range (*Figure 4*).

Finally, we examined the relationship between the connectivity patterns from the selected cortical parcels to the right cerebellum and the coupling of the speech envelope with the oscillations in the STG. The speech–brain coupling was computed using a multivariate mutual information (MI) approach presented in our recently published study (*Abbasi et al., 2023*). We investigated the correlations between the directional connectivity indices and MI values across participants for each parcel and frequency band in speaking condition. Our analysis showed significant negative correlations between the RCB6 to L_A5 connectivity and the speech–STG coupling in the theta band (at 130 ms lag) for speaking condition (*Figure 5*; p < 0.05).

## Discussion

The present study utilized multivariate Granger causality analysis to investigate the frequency-specific neural representations of the internal forward model during continuous speaking and listening. Our results revealed significant connectivity from bilateral higher-order cortical areas to bilateral auditory areas (STG and STS) in lower-frequency bands (up to beta) during speaking, and in the opposite direction in gamma frequencies. Notably, subcortical areas also contributed to the internal forward model, with directional communication observed from the right cerebellum lobule VI (R-CB6) to the left auditory area below 30 Hz. Conversely, in frequencies above 30 Hz, we observed reverse signaling from the left auditory areas to the right cerebellum.

### Directed connectivity in speech production and perception

In this study, we aimed to investigate directional connectivity in speech production and perception. Using the automatic meta-analysis provided by neurosynth.org, we included cortical and subcortical brain regions involved in these processes. Our multivariate Granger causality analysis suggests a potential role for predictive coding implemented through distinct frequency channels in the auditory–motor domain during both continuous speaking and listening. This analysis revealed stronger connectivity from higher-order brain regions (including the cerebellum) to the auditory area at slower frequencies (below 40 Hz). This connectivity pattern might reflect the influence of these regions' predictions on auditory processing. Conversely, the analysis showed stronger connectivity in the opposite direction, from the auditory area to these regions, at faster frequencies (above 40 Hz). This pattern could potentially be indicative of prediction error signals being relayed back for updating predictions. Our study builds upon our previous work, which demonstrated top–down signaling in low frequencies from higher-order cortical areas to STG and STS and the reverse pattern in higher frequencies. Our findings are also supported by previous studies showing distinct frequency channels for feedforward

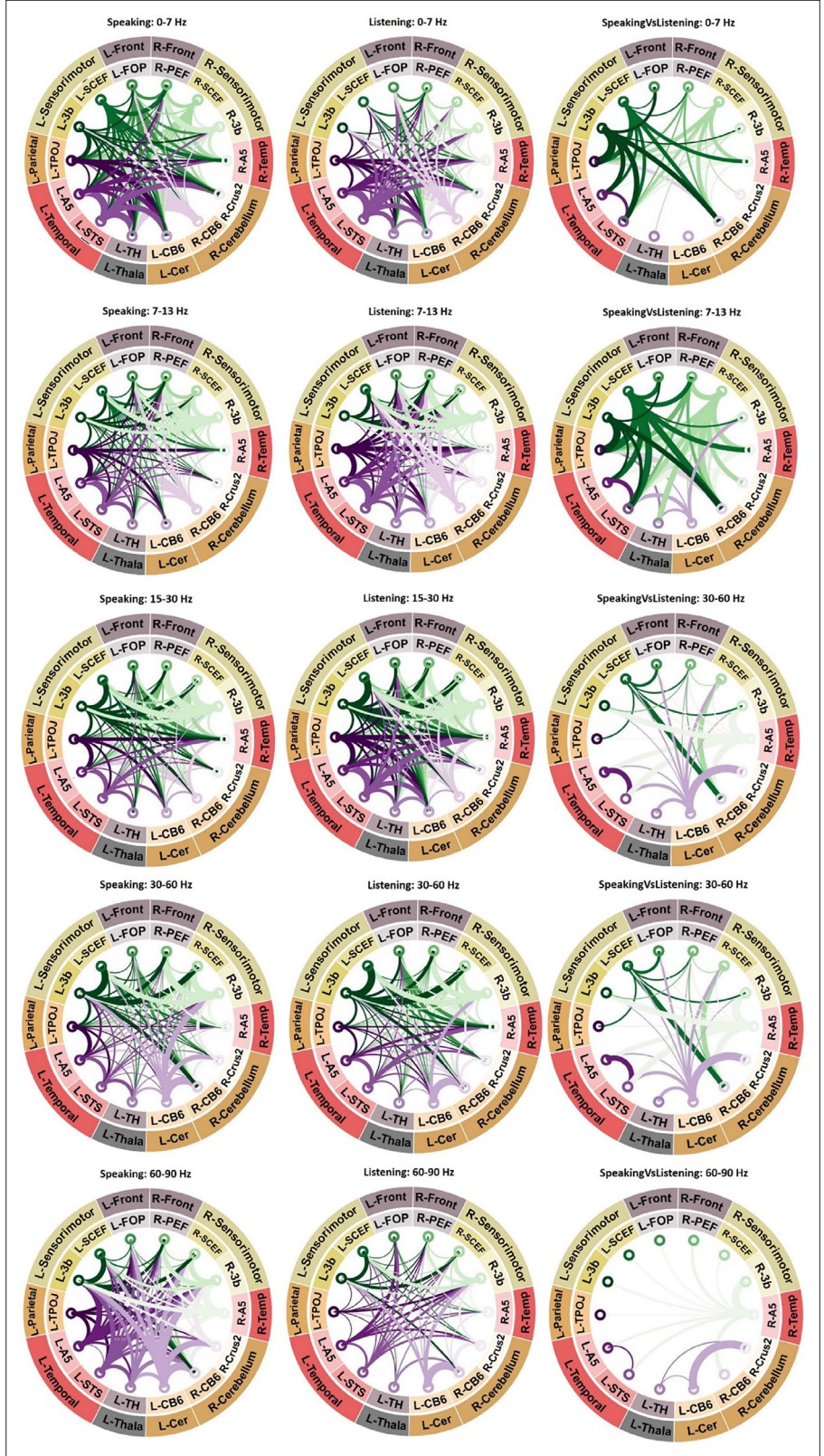

**Figure 3.** Connectivity analysis. Significant connectivity between 14 ROIs involved in speech production and perception. A cluster-based permutation test was used to detect significant connectivity patterns. The results of statistical analysis revealed significant connectivities between different brain areas during speaking (first column of connectograms), listening (second column of connectograms), and the comparison between speaking and

*Figure 3 continued on next page*

*Figure 3 continued*
listening conditions (third column of connectograms) across various frequency bands. In the connectograms, nodes represent the brain areas and edges represent the strength and direction of the connections between them. The thickness of the edges indicates the magnitude of the *t*-values, while color indicates the directionality of the connectivity. In other words, when node A connects to node B, the edge will have the same color as node A, and vice versa when node B connects to node A. Note that only significant connections are shown (p < 0.05, cluster correction).

and feedback communication between two hierarchically different auditory areas (*Fontolan et al., 2014*) and demonstrating that prediction errors are represented in gamma power while predictions are represented in lower-frequency beta power (*Sedley et al., 2016*).

Based on our meta-analysis results, several subcortical areas such as cerebellum and thalamus are involved in speech production and perception. Previous studies have supported the cerebellum's role in speech production and perception (*Giraud et al., 2007*; *Skipper and Lametti, 2021*). Our connectivity analysis findings demonstrate the involvement of the cerebellum in the implementation of predictive coding within speech networks. Specifically, our results indicate directed connectivity from the right cerebellum to left temporal areas (L-A5 and L-STS) in lower frequencies, and the reverse direction in higher frequencies, supporting the role of the cerebellum in predictive processing. Similar to the connections between higher-order cortical areas and temporal areas, feedback signaling between the cerebellum and temporal areas occurs in low frequencies (below 40 Hz), likely conveying timing information for upcoming speech. In contrast, feedforward signaling occurs in gamma frequency (above 60 Hz) from temporal areas to the cerebellum, facilitating the update of sensory predictions. Comparison between speech production and perception conditions revealed stronger feedback signaling from the cerebellum to temporal areas during speech production, aligning with the nature of updating sensory predictions in this context. Our findings closely align with recent studies that have also investigated the role of the cerebellum in predictive internal modeling mechanisms in speech production and perception (*Stockert et al., 2021*; *Todorović et al., 2024*). These studies suggest that the combined yet distinct activation of temporal, parietal, and cerebellar regions during internal and external monitoring points toward their involvement in auditory and somatosensory targets and continuous updating of auditory environmental models. Moreover, temporo-cerebellar coupling may underlie the precise encoding of temporal structure and support the ongoing optimization of spectro-temporal models of the auditory environment within a network comprising the prefrontal cortex, temporal cortex, and cerebellum (*Kotz et al., 2014*; *Stockert et al., 2021*; *Todorović et al., 2024*).

Our findings of significant negative correlations between right cerebellar connectivity to the left temporal area and speech–STG coupling in the theta band during speech production resonate with our prior work (*Abbasi et al., 2023*). In our previous study, we reported a similar negative correlation involving theta-range speech–brain coupling in the left auditory area and top–down beta connectivity from motor areas. These observations parallel existing research on sensory attenuation, where the brain predicts the sensory outcomes of self-generated actions, resulting in reduced cortical responses (*Martikainen et al., 2005*). In earlier studies, we associated this phenomenon with the modulation of beta-range directional couplings originating from motor cortices toward bilateral auditory regions, indicating predictive processes (*Abbasi and Gross, 2020*). Now, our new results introduce the cerebellum as another key node in this sensory attenuation mechanism, expanding our understanding of how the brain anticipates and minimizes sensory responses during speech production.

The similarities observed between the feedback and feedforward signaling from the cerebellum and higher-order cortical areas to the temporal areas suggest a shared contribution in predicting the sensory consequences of generated speech. It is proposed that cortico-cerebellar and cortico-cortical predictions interact in speech networks, with the cerebellum potentially involved in predicting well-learned speech, while the cortex flexibly applies predictions in novel contexts (*Ackermann, 2008*; *Ackermann et al., 1998*; *Skipper and Lametti, 2021*). Notably, the results of our previous findings (*Abbasi et al., 2023*) together with the current results presented in this study align with recent studies comparing speech production and perception, which highlight that these processes may rely on the same neural representations but are differentiated in their temporal dynamics (*Giglio et al., 2024*; *Fairs et al., 2021*). Moreover, the current findings deepen our understanding of the role of

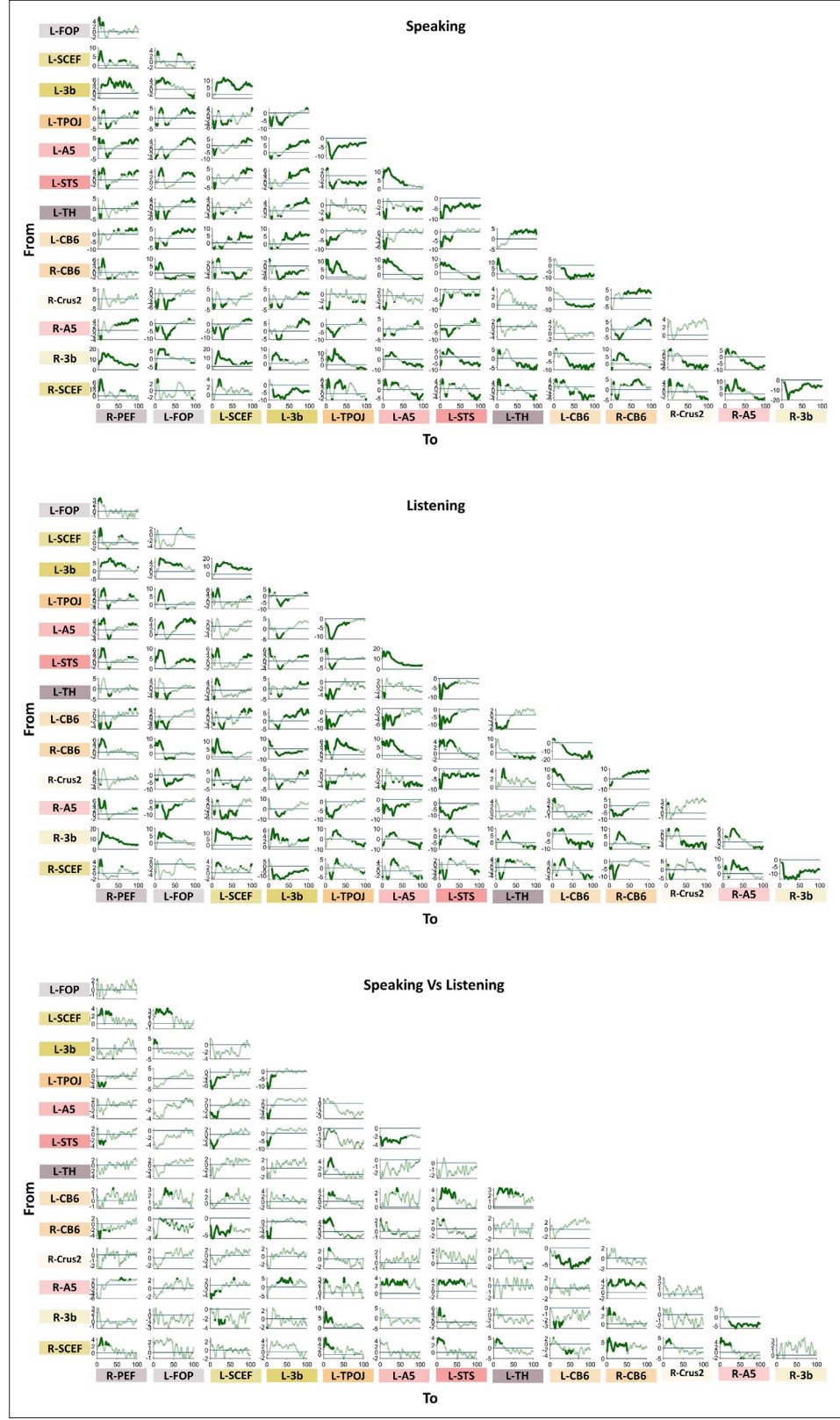

**Figure 4.** Connectivity results. Spectrally resolved directed asymmetry index (DAI) between all pairs of ROIs. A cluster-based permutation test was used to detect significant connectivity patterns. The results of statistical analysis revealed significant connectivities between different brain areas during speaking (top), listening (middle), and the comparison between speaking and listening conditions (bottom). The directionality is from *y*- to *x*-axis.

*Figure 4 continued on next page*

*Figure 4 continued*

Note that significant values are highlighted with increased line width (p < 0.05, cluster correction). *x*- and *y*-axes represent frequency (Hz) and *t*-values, respectively.

the cerebellum in speech production and perception and align with previous research highlighting its involvement in predictive processing mechanisms.

We also observed significant connectivity between the right cerebellum and the left parietal cortex, specifically in the low-frequency ranges, which peaked in the alpha range. This result is consistent with previous research demonstrating the alpha band connectivity between parietal and temporal areas during speech production and perception (*Abbasi et al., 2023*). We and others previously suggested that the parietal cortex might control and modulate the alpha rhythms in the early auditory areas which serves as a multidimensional filter (*Abbasi et al., 2023*; *Lakatos et al., 2019*). Moreover, our current results reveal that during speaking and listening, the parietal cortex not only receives inputs from higher cortical areas such as the motor cortex, but also receives inputs from the cerebellum. These inputs enable the parietal cortex to encode predicted sensory consequences during speaking and provide top–down signals to early auditory areas. Additionally, our new findings demonstrate stronger connectivity from the cerebellum to the parietal areas during the speaking condition compared to the listening condition. This enhanced connectivity may be attributed to the selective inhibition of auditory signals during speaking, as predicted by the internal predictive coding model (*Cao et al., 2017*; *Floegel et al., 2020*).

The thalamus is another subcortical structure involved in speech perception and production which appears to play a significant role as it serves as a crucial intermediary in the cortico-subcortical neural network involving the cerebellum (*Barbas et al., 2013*; *Silveri, 2021*). Our findings reinforce this notion, revealing significant connectivity from the right cerebellum to the left thalamus in the low-frequency band, with the opposite direction observed in the high-frequency band during speaking. These connectivity patterns align with the results observed between the right cerebellum and other cortical areas, providing further support for the thalamus's role in interconnecting cortical and subcortical structures, facilitating communication and coordination in various stages of verbal production (*Crosson, 2013*). It is important to acknowledge that, while MEG's spatial resolution is inherently lower for deep brain regions like the cerebellum compared to cortical areas, there is a large body of evidence (such as *Attal and Schwartz, 2013*) demonstrating its capability to record signals from these regions, including the thalamus and cerebellum.

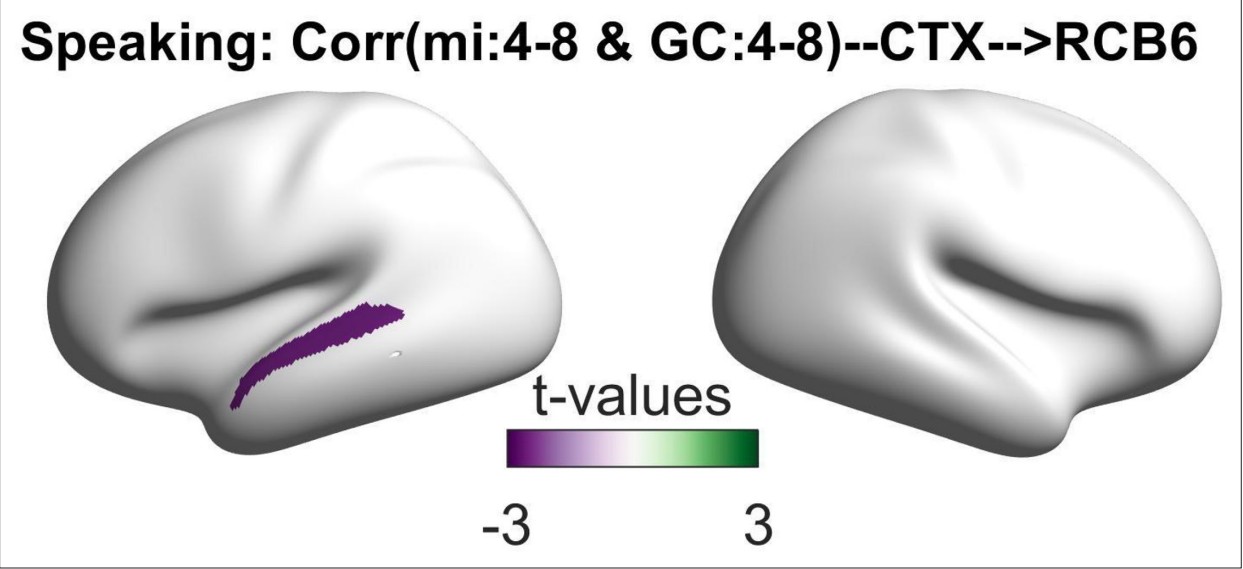

**Figure 5.** Correlation between speech–brain coupling and Granger causality. This plot depicts the relationship between speech–brain coupling in the left superior temporal gyrus (STG) at theta frequency (4–8 Hz) with a positive lag of 130 ms and (right cerebellum to left STG) Granger causality (GC) during the speaking condition. The negative correlation indicates that weaker speech–brain coupling in the theta band is associated with stronger directional information flow from the right cerebellar lobule VI to the left temporal areas in the theta frequency band.

Our latest findings underscore that both speech production and perception involve distinct frequency channels for predictive processing, emphasizing the role of feedback and feedforward signaling in supporting this aspect of the predictive coding framework. Notably, we have also unveiled the anticipated role of the cerebellum within this framework. However, definitive conclusions should be drawn with caution given recent studies raising concerns about the notion that top–down and bottom–up signals can only be transmitted via separate frequency channels (*Ferro et al., 2021*; *Schneider et al., 2021*; *Vinck et al., 2023*). Therefore, further investigation is warranted to thoroughly assess the alignment between our results and the contribution of the predictive coding framework in speech production and perception.

## Methods

### Participants

Thirty native German-speaking participants (15 males, mean age 25.1 ± 2.8 years [*M* ± SD], range 20–32 years) were recruited for this study. Prior written informed consent was obtained before measurements and participants received monetary compensation after partaking. The study was approved by the local ethics committee and conducted in accordance with the Declaration of Helsinki. This study additionally re-analyzes data previously collected for a study published in *Abbasi et al., 2023*.

### Recordings

MEG, EMG, and speech signals were recorded simultaneously. The speech recording had a sampling rate of 44.1 kHz, whereas the MEG, a 275-channel, whole-head sensor system (OMEGA 275, VSM Medtech Ltd, Vancouver, BC, Canada) was sampled with 1200 Hz.

In order not to cause any artifacts by the microphone used for capturing audio data, it was placed at a distance of 155 cm from the participants mouth. Three pairs of EMG surface electrodes were placed after tactile inspection to find the correct location to capture muscle activity from the m. genioglossus, m. orbicularis oris, and m. zygomaticus major (for exact location see Figure 1 in *Abbasi et al., 2021*). One pair of electrodes was used for each muscle with about 1 cm between electrodes. A low-pass online filter with a 300-Hz cut-off was applied to the recorded MEG and EMG data.

### Paradigm

Participants were asked to sit relaxed while performing the given tasks and to keep their eyes on a white fixation cross. The experiment was split in two separate parts: The first one consisted of answering given questions, each for 60 s, thus recording overt speech. Participants had to answer seven questions covering neutral topics, such as 'What does a typical weekend look like for you?'. A color change from white to blue fixation cross indicated the beginning of the time period in which participants should speak and the end was marked by a color change back to white. The second part focused on perceiving speech in the way that participants listened to their own answers from part one. The list of questions as well as further details of the paradigm presented can be found in *Abbasi et al., 2023*.

### Preprocessing and data analysis

Prior to data analysis, MEG data were visually inspected. No jump artifacts or bad channels were detected. A discrete Fourier transform filter was applied to eliminate 50 Hz line noise from the continuous MEG and EMG data. Moreover, EMG data were highpass-filtered at 20 Hz and rectified. Continuous head position and rotation were extracted from the fiducial coils placed at anatomical landmarks (nasion, left, and right ear canals). MEG, EMG, and head movement signals were downsampled to 256 Hz and segmented to non-overlapping 60 s trials corresponding to each of their overt answers. In the preprocessing and data analysis steps, custom-made scripts in Matlab R2020 (The Mathworks, Natick, MA, USA) in combination with the Matlab-based FieldTrip toolbox (*Oostenveld et al., 2011*) were used in accord with current MEG guidelines (*Gross et al., 2013a*).

### Artifact rejection

For removing the speech-related artifacts we used the pipeline presented in *Abbasi et al., 2021*. In a nutshell, the artifact rejection comprises four major steps: (1) Head movement-related artifact

was initially reduced by incorporating the head position time-series into the general linear model using regression analysis (*Stolk et al., 2013*). (2) To further remove the residual artifact, SVD was used to estimate the spatial subspace (components) containing the speech-related artifact from the MEG data. (3) Artifactual components were detected via visual inspections and MI analysis and then removed from the single-trial data (*Abbasi et al., 2016*). (4) Finally, all remaining components were back-transformed to the sensor level.

## Source localization

For source localization we aligned individual T1-weighted anatomical MRI scans with the digitized head shapes using the iterative closest point algorithm. Then, we segmented the MRI scans and generated single-shell volume conductor models (*Nolte, 2003*), and used this to create forward models. For group analyses, individual MRIs were linearly transformed to an MNI template provided by Fieldtrip. Next, the linearly constrained minimum variance algorithm was used to compute time-series for each voxel on a 5-mm grid. The time-series were extracted for each dipole orientation, resulting in three time-series per voxel. The reduced version of the HCP brain atlas as well as AAL atlas were applied on the source space time-series in order to reduce the dimensionality of the data, resulting in 230 cortical parcels (*Tait et al., 2020*) and 116 subcortical parcels, respectively. Since the HCP atlas only covers cortical areas we used the AAL atlas for subcortical areas. Finally, we extracted the first three components of an SVD of time-series from all dipoles in this parcel, explaining most of the variance.

## ROI selection

In this study, we focused on the brain regions that are involved in speech production and perception. In order to find the areas involved in speech production and perception network, we utilized the automatic meta-analysis provided by neurosynth.org using the term 'speech' that resulted in a meta-analysis of 642 fMRI studies highlighting the brain areas involved in speech production and perception (*Yarkoni et al., 2011*). We extracted the MNI coordinates of active areas from the presented map, found their corresponding voxels and identified the respective parcels on HCP and AAL atlases where these voxels are located. This resulted in 14 cortical and subcortical parcels (see *Figure 1* and *Table 1*).

## Connectivity analysis

We performed connectivity analysis by using an mGC approach (*Schaum et al., 2021*; *Dhamala et al., 2008*). We computed the mGC to determine the directionality of functional coupling between all the detected involved parcels, in pairwise steps, during speech production and perception. Initially, the source signals were divided into trials of 4 s, with 500 ms overlap. We used the fast Fourier transform in combination with multitapers (2 Hz smoothing) to compute the cross-spectral density matrix of the

**Table 1.** Selected ROI labels from HCP and AAL atlases.

| 1  | L-FOP       | Left frontal opercular area                 |
|----|-------------|---------------------------------------------|
| 2  | R-PEF+6v    | Right premotor eyefield + ventral area 6    |
| 3  | L-SCEF      | Left supplementary and cingulate eyefield   |
| 4  | R-SCEF      | Right supplementary and cingulate eyefield  |
| 5  | L-3b        | Left primary somatosensory cortex           |
| 6  | R-3b        | Right primary somatosensory cortex          |
| 7  | L-STS       | Left superior temporal sulcus               |
| 8  | L-TPOJ1     | Left TemporoParietoOccipital Junction 1     |
| 9  | L-A5        | Left auditory complex 5                      |
| 10 | R-A5        | Right auditory complex 5                     |
| 11 | L-TH        | Left thalamus                                |
| 12 | R-CB-Cruss2 | Right cerebellum crus 2                       |
| 13 | L-CB-6      | Left cerebellum 6                            |
| 14 | R-CB-6      | Right cerebellum 6                           |

trials. Next, using a blockwise approach, we considered the first three SVD components of each parcel as a block and estimated the connectivity between STG and other parcels. Finally, we computed the directed influence asymmetry index (DAI) defined by *Bastos et al., 2015* as

$$DAI = \frac{\left[ mGC\left(parcel \rightarrow STG\right) - mGC\left(STG \rightarrow parcel\right) \right]}{\left[ mGC\left(parcel \rightarrow STG\right) + mGC\left(STG \rightarrow parcel\right) \right]}$$

Therefore, a positive DAI for a given frequency indicates that the selected parcel conveys feedforward influences to STG in this frequency, and a negative DAI indicates feedback influences. Note that for the connectivity analysis, we used MEG data with 1200 Hz sampling rate without downsampling.

## Statistical analysis

We determined significant connectivity patterns (DAI values) in both speaking and listening conditions using nonparametric cluster-based permutation tests (*Maris and Oostenveld, 2007*). First, we estimated the statistical contrast of connectivities during speaking and listening compared to zero for each parcel and participant. Second, the DAI values in the speaking condition were contrasted with DAI values in the listening condition at the group level. The statistical analysis was conducted for different frequency bands (Delta/Theta (0–7 Hz), alpha (7–13 Hz), beta (15–30 Hz), gamma (30–60 Hz), and high gamma (60–90 Hz)) using a dependent-samples *t*-test. We used a cluster-based correction to account for multiple comparisons across frequencies and parcels. We performed 5000 permutations and set the critical alpha value at 0.05.

## Speech–brain coupling

For each parcel, we calculated the complex-valued spectral estimates of the three SVD components using multitaper analysis with ±2 Hz spectral smoothing on 2-s windows with 50% overlap. Subsequently, we estimated the MI between the speech envelope and the combined spectral information from all three time-series using Gaussian Copula MI. This approach resulted in a single MI value per parcel, reflecting speech–brain coupling specifically at the chosen 130 ms lag.

## Correlation analysis

To examine the potential relationship between our connectivity findings from all the parcels to R-CB6 and speech–STG coupling, we conducted nonparametric cluster-based permutation tests. Our first step was to calculate top–down connectivity values for selected parcels and frequency bands, then to compute speech–STG couplings for each frequency band. To account for multiple comparisons across parcels, we employed the Pearson method implemented in the ft_statfun_correlationT function in Fieldtrip, with cluster-based correction. Our analysis was repeated for different frequency bands. Therefore, our results are not corrected across frequencies. We performed 5000 permutations and set the critical alpha value at 0.05.

## Acknowledgements

We acknowledge support by the Open Access Publication Fund of University of Münster. JG was also supported by the DFG (GR 2024/5-1, GR 2024/11-1, and GR 2024/12-1).

## Additional information

### Funding

| Funder | Grant reference number | Author |
|---|---|---|
| Deutsche Forschungsgemeinschaft | GR 2024/5-1 | Joachim Gross |
| Deutsche Forschungsgemeinschaft | GR 2024/11-1 | Joachim Gross |

| Funder | Grant reference number | Author |
|---|---|---|
| Deutsche Forschungsgemeinschaft | GR 2024/12-1 | Joachim Gross |

The funders had no role in study design, data collection, and interpretation, or the decision to submit the work for publication.

## Author contributions

Omid Abbasi, Formal analysis, Validation, Investigation, Visualization, Methodology, Writing – original draft, Project administration, Writing – review and editing; Nadine Steingräber, Data curation, Methodology, Writing – original draft; Nikos Chalas, Data curation, Writing – original draft; Daniel S Kluger, Visualization, Writing – original draft; Joachim Gross, Conceptualization, Supervision, Funding acquisition, Investigation, Methodology, Writing – original draft, Project administration, Writing – review and editing

## Author ORCIDs

Omid Abbasi ⓘ https://orcid.org/0000-0003-2169-8498
Daniel S Kluger ⓘ https://orcid.org/0000-0002-0691-794X
Joachim Gross ⓘ https://orcid.org/0000-0002-3994-1006

## Ethics

Prior written informed consent was obtained before measurements and participants received monetary compensation after partaking. The study was approved by the local ethics committee and conducted in accordance with the Declaration of Helsinki.

Reviewer #1 (Public Review): https://doi.org/10.7554/eLife.97083.2.sa1
Reviewer #2 (Public Review): https://doi.org/10.7554/eLife.97083.2.sa2
Reviewer #3 (Public Review): https://doi.org/10.7554/eLife.97083.2.sa3
Author response https://doi.org/10.7554/eLife.97083.2.sa4

# Additional files

## Supplementary files
• MDAR checklist

## Data availability

Custom-made scripts used for data analysis are publicly available at link: https://osf.io/9fq47/. Additionally, an example dataset is included at the same repository to facilitate understanding of the data structure. The raw data contains participants' audio speech files and is protected by data privacy laws, preventing public sharing. However, we are committed to responsible data sharing and can make the raw data available upon reasonable request, subject to data privacy regulations. To access the raw data, interested researchers can contact the corresponding author of this manuscript by informal email. They only need to briefly explain their research goals and the specific data subsets they require. No formal proposal is required.

The following dataset was generated:

| Author(s) | Year | Dataset title | Dataset URL | Database and Identifier |
|---|---|---|---|---|
| Abbasi O | 2022 | Preprocessed and anonymised data set example as well as source codes of speech perception and production study! | https://osf.io/9fq47/ | Open Science Framework, 9fq47 |

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
