## [Editor Report · eLife assessment]

Abbasi and colleagues use Granger causality to explore the cortico-subcortical dynamics during speaking and listening. They find **valuable** evidence for bi-directional connectivity in distinct frequency bands as a function of behaviour, but currently offer **incomplete** support for the validity of their analyses and the predictive coding interpretation of their results.

---

## [Referee Report · Reviewer #1 (Public Review)]

Abbasi et al. assess in this MEG study the directed connectivity of both cortical and subcortical regions during continuous speech production and perception. The authors observed bidirectional connectivity patterns between speech-related cortical areas as well as subcortical areas in production and perception. Interestingly, they found in speaking low-frequency connectivity from subcortical (the right cerebellum) to cortical (left superior temporal) areas, while connectivity from the cortical to subcortical areas was in the high frequencies. In listening a similar cortico-subcortical connectivity pattern was observed for the low frequencies, but the reversed connectivity in the higher frequencies was absent.

The work by Abbasi and colleagues addresses a relevant, novel topic, namely understanding the brain dynamics between speaking and listening. This is important because traditionally production and perception of speech and language are investigated in a modality-specific manner. To have a more complete understanding of the neurobiology underlying these different speech behaviors, it is key to also understand their similarities and differences. Furthermore, to do so, the authors utilize state-of-the-art directed connectivity analyses on MEG measurements, providing a quite detailed profile of cortical and subcortical interactions for the production and perception of speech. Importantly, and perhaps most interesting in my opinion, is that the authors find evidence for frequency-specific directed connectivity, which is (partially) different between speaking and listening. This could suggest that both speech behaviors rely (to some extent) on similar cortico-cortical and cortico-subcortical networks, but different frequency-specific dynamics.

These elements mentioned above (investigation of both production and perception, both cortico-cortical and cortico-subcortical connectivity is considered, and observing frequency-specific connectivity profiles within and between speech behaviors), make for important novel contributions to the field. Notwithstanding these strengths, I find that they are especially centered on methodology and functional anatomical description, but that precise theoretical contributions for neurobiological and cognitive models of speech are less transparent. This is in part because the study compares speech production and perception in general, but no psychophysical or psycholinguistic manipulations are considered. I also have some critical questions about the design which may pose some confounds in interpreting the data, especially with regard to comparing production and perception.

(1) While the cortico-cortical and cortico-subcortical connectivity profiles highlighted in this study and the depth of the analyses are impressive, what these data mean for models of speech processing remains on the surface. This is in part due, I believe, to the fact that the authors have decided to explore speaking and listening in general, without targeting specific manipulations that help elucidate which aspects of speech processing are relevant for the particular connectivity profiles they have uncovered. For example, the frequency-specific directed connectivity is it driven by low-level psychophysical attributes of the speech or by more cognitive linguistic properties? Does it relate to the monitoring of speech, timing information, and updating of sensory predictions? Without manipulations trying to target one or several of these components, as some of the referenced work has done (e.g., Floegel et al., 2020; Stockert et al., 2021; Todorović et al., 2023), it is difficult to draw concrete conclusions as to which representations and/or processes of speech are reflected by the connectivity profiles. An additional disadvantage of not having manipulations within each speech behavior is that it makes the comparison between listening and speaking harder. That is, speaking and listening have marked input-output differences which likely will dominate any comparison between them. These physically driven differences (or similarities for that matter; see below) can be strongly reduced by instead exploring the same manipulations/variables between speaking and listening. If possible (if not to consider for future work), it may be interesting to score psychophysical (e.g., acoustic properties) or psycholinguistic (e.g., lexical frequency) information of the speech and see whether and how the frequency-specific connectivity profiles are affected by it.

(2) Recent studies comparing the production and perception of language may be relevant to the current study and add some theoretical weight since their data and interpretations for the comparisons between production and perception fit quite well with the observations in the current work. These studies highlight that language processes between production and perception, specifically lexical and phonetic processing (Fairs et al., 2021), and syntactic processing (Giglio et al., 2024), may rely on the same neural representations, but are differentiated in their (temporal) dynamics upon those shared representations. This is relevant because it dispenses with the classical notion in neurobiological models of language where production and perception rely on (partially) dissociable networks (e.g., Price, 2010). Rather those data suggest shared networks where different language behaviors are dissociated in their dynamics. The speech results in this study nicely fit and extend those studies and their theoretical implications.

(3) The authors align the frequency-selective connectivity between the right cerebellum and left temporal speech areas with recent studies demonstrating a role for the right cerebellum for the internal modelling in speech production and monitoring (e.g., Stockert et al., 2021; Todorović et al., 2023). This link is indeed interesting, but it does seem relevant to point out that at a more specific scale, it does not concern the exact same regions between those studies and the current study. That is, in the current study the frequency-specific connectivity with temporal regions concerns lobule VI in the right cerebellum, while in the referenced work it concerns Crus I/II. The distinction seems relevant since Crus I/II has been linked to the internal modelling of more cognitive behavior, while lobule VI seems more motor-related and/or contextual-related (e.g., D'Mello et al., 2020; Runnqvist et al., 2021; Runnqvist, 2023).

(4) On the methodological side, my main concern is that for the listening condition, the authors have chosen to play back the speech produced by the participants in the production condition. Both the fixed order as well as hearing one's own speech as listening condition may produce confounds in data interpretation, especially with regard to the comparison between speech production and perception. Could order effects impact the observed connectivity profiles, and how would this impact the comparison between speaking and listening? In particular, I am thinking of repetition effects present in the listening condition as well as prediction, which will be much more elevated for the listening condition than the speaking condition. The fact that it also concerns their own voice furthermore adds to the possible predictability confound (e.g., Heinks-Maldonado et al., 2005). In addition, listening to one's speech which just before has been articulated may, potentially strategically even, enhance inner speech and "mouthing" in the participants, hereby thus engaging the production mechanism. Similarly, during production, the participants already hear their own voice (which serves as input in the subsequent listening condition). Taken together, both similarities or differences between speaking and listening connectivity may have been due to or influenced by these order effects, and the fact that the different speech behaviors are to some extent present in both conditions.

(5) The ability of the authors to analyze the spatiotemporal dynamics during continuous speech is a potentially important feat of this study, given that one of the reasons that speech production is much less investigated compared to perception concerns motor and movement artifacts due to articulation (e.g., Strijkers et al., 2010). Two questions did spring to mind when reading the authors' articulation artifact correction procedure: If I understood correctly, the approach comes from Abbasi et al. (2021) and is based on signal space projection (SSP) as used for eye movement corrections, which the authors successfully applied to speech production. However, in that study, it concerned the repeated production of three syllables, while here it concerns continuous speech of full words embedded in discourse. The articulation and muscular variance will be much higher in the current study compared to three syllables (or compared to eye movements which produce much more stable movement potentials compared to an entire discourse). Given this, I can imagine that corrections of the signal in the speaking condition were likely substantial and one may wonder (1) how much signal relevant to speech production behavior is lost?; (2) similar corrections are not necessary for perception, so how would this marked difference in signal processing affect the comparability between the modalities?

References:

- Abbasi, O., Steingräber, N., & Gross, J. (2021). Correcting MEG artifacts caused by overt speech. Frontiers in Neuroscience, 15, 682419.

- D'Mello, A. M., Gabrieli, J. D., & Nee, D. E. (2020). Evidence for hierarchical cognitive control in the human cerebellum. Current Biology, 30(10), 1881-1892.

- Fairs, A., Michelas, A., Dufour, S., & Strijkers, K. (2021). The same ultra-rapid parallel brain dynamics underpin the production and perception of speech. Cerebral Cortex Communications, 2(3), tgab040.

- Floegel, M., Fuchs, S., & Kell, C. A. (2020). Differential contributions of the two cerebral hemispheres to temporal and spectral speech feedback control. Nature Communications, 11(1), 2839.

- Giglio, L., Ostarek, M., Sharoh, D., & Hagoort, P. (2024). Diverging neural dynamics for syntactic structure building in naturalistic speaking and listening. Proceedings of the National Academy of Sciences, 121(11), e2310766121.

- Heinks‐Maldonado, T. H., Mathalon, D. H., Gray, M., & Ford, J. M. (2005). Fine‐tuning of auditory cortex during speech production. Psychophysiology, 42(2), 180-190.

- Price, C. J. (2010). The anatomy of language: a review of 100 fMRI studies published in 2009. Annals of the new York Academy of Sciences, 1191(1), 62-88.

- Runnqvist, E., Chanoine, V., Strijkers, K., Pattamadilok, C., Bonnard, M., Nazarian, B., ... & Alario, F. X. (2021). Cerebellar and cortical correlates of internal and external speech error monitoring. Cerebral Cortex Communications, 2(2), tgab038.

- Runnqvist, E. (2023). Self-monitoring: The neurocognitive basis of error monitoring in language production. In Language production (pp. 168-190). Routledge.

- Stockert, A., Schwartze, M., Poeppel, D., Anwander, A., & Kotz, S. A. (2021). Temporo-cerebellar connectivity underlies timing constraints in audition. Elife, 10, e67303.

- Strijkers, K., Costa, A., & Thierry, G. (2010). Tracking lexical access in speech production: electrophysiological correlates of word frequency and cognate effects. Cerebral cortex, 20(4), 912-928.

- Todorović, S., Anton, J. L., Sein, J., Nazarian, B., Chanoine, V., Rauchbauer, B., ... & Runnqvist, E. (2023). Cortico-cerebellar monitoring of speech sequence production. Neurobiology of Language, 1-21.

---

## [Referee Report · Reviewer #2 (Public Review)]

Summary:

The authors re-analyse MEG data from a speech production and perception study and extend their previous Granger causality analysis to a larger number of cortical-cortical and in particular cortical-subcortical connections. Regions of interest were defined by means of a meta-analysis using Neurosynth.org and connectivity patterns were determined by calculating directed influence asymmetry indices from the Granger causality analysis results for each pair of brain regions. Abbasi et al. report feedforward signals communicated via fast rhythms and feedback signals via slow rhythms below 40 Hz, particularly during speaking. The authors highlight one of these connections between the right cerebellum lobule VI and auditory association area A5, where in addition the connection strength correlates negatively with the strength of speech tracking in the theta band during speaking (significant before multiple comparison correction). Results are interpreted within a framework of active inference by minimising prediction errors.

While I find investigating the role of cortical-subcortical connections in speech production and perception interesting and relevant to the field, I am not yet convinced that the methods employed are fully suitable to this endeavour or that the results provide sufficient evidence to make the strong claim of dissociation of bottom-up and top-down information flow during speaking in distinct frequency bands.

Strengths:

The investigation of electrophysiological cortical-subcortical connections in speech production and perception is interesting and relevant to the field. The authors analyse a valuable dataset, where they spent a considerable amount of effort to correct for speech production-related artefacts. Overall, the manuscript is well-written and clearly structured.

Weaknesses:

The description of the multivariate Granger causality analysis did not allow me to fully grasp how the analysis was performed and I hence struggled to evaluate its appropriateness.

Knowing that (1) filtered Granger causality is prone to false positives and (2) recent work demonstrates that significant Granger causality can simply arise from frequency-specific activity being present in the source but not the target area without functional relevance for communication (Schneider et al. 2021) raises doubts about the validity of the results, in particular with respect to their frequency specificity. These doubts are reinforced by what I perceive as an overemphasis on results that support the assumption of specific frequencies for feedforward and top-down connections, while findings not aligning with this hypothesis appear to be underreported. Furthermore, the authors report some main findings that I found difficult to reconcile with the data presented in the figures. Overall, I feel the conclusions with respect to frequency-specific bottom-up and top-down information flow need to be moderated and that some of the reported findings need to be checked and if necessary corrected.

Major points

(1) I think more details on the multivariate GC approach are needed. I found the reference to Schaum et al., 2021 not sufficient to understand what has been done in this paper. Some questions that remained for me are:

(i) Does multivariate here refer to the use of the authors' three components per parcel or to the conditioning on the remaining twelve sources? I think the latter is implied when citing Schaum et al., but I'm not sure this is what was done here?

If it was not: how can we account for spurious results based on indirect effects?

(ii) Did the authors check whether the GC of the course-target pairs was reliably above the bias level (as Schaum et. al. did for each condition separately)? If not, can they argue why they think that their results would still be valid? Does it make sense to compute DAIs on connections that were below the bias level? Should the data be re-analysed to take this concern into account?

(iii) You may consider citing the paper that introduced the non-parametric GC analysis (which Schaum et al. then went on to apply): Dhamala M, Rangarajan G, Ding M. Analyzing Information Flow in Brain Networks with Nonparametric Granger Causality. Neuroimage. 2008; 41(2):354-362. https://doi.org/10.1016/j.neuroimage.2008.02. 020

(2) GC has been discouraged for filtered data as it gives rise to false positives due to phase distortions and the ineffectiveness of filtering in the information-theoretic setting as reducing the power of a signal does not reduce the information contained in it (Florin et al., 2010; Barnett and Seth, 2011; Weber et al. 2017; Pinzuti et al., 2020 - who also suggest an approach that would circumvent those filter-related issues). With this in mind, I am wondering whether the strong frequency-specific claims in this work still hold.

(3) I found it difficult to reconcile some statements in the manuscript with the data presented in the figures:

(i) Most notably, the considerable number of feedforward connections from A5 and STS that project to areas further up the hierarchy at slower rhythms (e.g. L-A5 to R-PEF, R-Crus2, L CB6 L-Tha, L-FOP and L-STS to R-PEF, L-FOP, L-TOPJ or R-A5 as well as R-STS both to R-Crus2, L-CB6, L-Th) contradict the authors' main message that 'feedback signals were communicated via slow rhythms below 40 Hz, whereas feedforward signals were communicated via faster rhythms'. I struggled to recognise a principled approach that determined which connections were highlighted and reported and which ones were not.

(ii) "Our analysis also revealed robust connectivity between the right cerebellum and the left parietal cortex, evident in both speaking and listening conditions, with stronger connectivity observed during speaking. Notably, Figure 4 depicts a prominent frequency peak in the alpha band, illustrating the specific frequency range through which information flows from the cerebellum to the parietal areas." There are two peaks discernible in Figure 4, one notably lower than the alpha band (rather theta or even delta), the other at around 30 Hz. Nevertheless, the authors report and discuss a peak in the alpha band.

(iii) In the abstract: "Notably, high-frequency connectivity was absent during the listening condition." and p.9 "In contrast with what we reported for the speaking condition, during listening, there is only a significant connectivity in low frequency to the left temporal area but not a reverse connection in the high frequencies."

While Fig. 4 shows significant connectivity from R-CB6 to A5 in the gamma frequency range for the speaking, but not for the listening condition, interpreting comparisons between two effects without directly comparing them is a common statistical mistake (Makin and Orban de Xivry). The spectrally-resolved connectivity in the two conditions actually look remarkably similar and I would thus refrain from highlighting this statement and indicate clearly that there were no significant differences between the two conditions.

(iv) "This result indicates that in low frequencies, the sensory-motor area and cerebellum predominantly transmit information, while in higher frequencies, they are more involved in receiving it."

I don't think that this statement holds in its generality: L-CB6 and R-3b both show strong output at high frequencies, particularly in the speaking condition. While they seem to transmit information mainly to areas outside A5 and STS these effects are strong and should be discussed.

(4) "However, definitive conclusions should be drawn with caution given recent studies raising concerns about the notion that top-down and bottom-up signals can only be transmitted via separate frequency channels (Ferro et al., 2021; Schneider et al., 2021; Vinck et al., 2023)."

I appreciate this note of caution and think it would be useful if it were spelled out to the reader why this is the case so that they would be better able to grasp the main concerns here. For example, Schneider et al. make a strong point that we expect to find Granger-causality with a peak in a specific frequency band for areas that are anatomically connected when the sending area shows stronger activity in that band than the receiving one, simply because of the coherence of a signal with its own linear projection onto the other area. The direction of a Granger causal connection would in that case only indicate that one area shows stronger activity than the other in the given frequency band. I am wondering to what degree the reported connectivity pattern can be traced back to regional differences in frequency-specific source strength or to differences in source strength across the two conditions.

---

## [Referee Report · Reviewer #3 (Public Review)]

In the current paper, Abbasi et al. aimed to characterize and compare the patterns of functional connectivity across frequency bands (1 Hz - 90 Hz) between regions of a speech network derived from an online meta-analysis tool (Neurosynth.org) during speech production and perception. The authors present evidence for complex neural dynamics from which they highlight directional connectivity from the right cerebellum to left superior temporal areas in lower frequency bands (up to beta) and between the same regions in the opposite direction in the (lower) high gamma range (60-90 Hz). Abbasi et al. interpret their findings within the predictive coding framework, with the cerebellum and other "higher-order" (motor) regions transmitting top-down sensory predictions to "lower-order" (sensory) regions in the lower frequencies and prediction errors flowing in the opposite direction (i.e., bottom-up) from those sensory regions in the gamma band. They also report a negative correlation between the strength of this top-down functional connectivity and the alignment of superior temporal regions to the syllable rate of one's speech.

Strengths:

(1) The comprehensive characterization of functional connectivity during speaking and listening to speech may be valuable as a first step toward understanding the neural dynamics involved.

(2) The inclusion of subcortical regions and connectivity profiles up to 90Hz using MEG is interesting and relatively novel.

(3) The analysis pipeline is generally adequate for the exploratory nature of the work.

Weaknesses:

(1) The work is framed as a test of the predictive coding theory as it applies to speech production and perception, but the methodological approach is not suited to this endeavor.

(2) Because of their theoretical framework, the authors readily attribute roles or hierarchy to brain regions (e.g., higher- vs lower-order) and cognitive functions to observed connectivity patterns (e.g., feedforward vs feedback, predictions vs prediction errors) that cannot be determined from the data. Thus, many of the authors' claims are unsupported.

(3) The authors' theoretical stance seems to influence the presentation of the results, which may inadvertently misrepresent the (otherwise perfectly valid; cf. Abbasi et al., 2023) exploratory nature of the study. Thus, results about specific regions are often highlighted in figures (e.g., Figure 2 top row) and text without clear reasons.

(4) Some of the key findings (e.g., connectivity in opposite directions in distinct frequency bands) feature in a previous publication and are, therefore, interesting but not novel.

(5) The quantitative comparison between speech production and perception is interesting but insufficiently motivated.

(6) Details about the Neurosynth meta-analysis and subsequent selection of brain regions for the functional connectivity analyses are incomplete. Moreover, the use of the term 'Speech' in Neurosynth seems inappropriate (i.e., includes irrelevant works, yielding questionable results). The approach of using separate meta-analyses for 'Speech production' and 'Speech perception' taken by Abbasi et al. (2023) seems more principled. This approach would result, for example, in the inclusion of brain areas such as M1 and the BG that are relevant for speech production.

(7) The results involving subcortical regions are central to the paper, but no steps are taken to address the challenges involved in the analysis of subcortical activity using MEG. Additional methodological detail and analyses would be required to make these results more compelling. For example, it would be important to know what the coverage of the MEG system is, what head model was used for the source localization of cerebellar activity, and if specific preprocessing or additional analyses were performed to ensure that the localized subcortical activity (in particular) is valid.

(8) The results and methods are often detailed with important omissions (a speech-brain coupling analysis section is missing) and imprecisions (e.g., re: Figure 5; the Connectivity Analysis section is copy-pasted from their previous work), which makes it difficult to understand what is being examined and how. (It is also not good practice to refer the reader to previous publications for basic methodological details, for example, about the experimental paradigm and key analyses.) Conversely, some methodological details are given, e.g., the acquisition of EMG data, without further explanation of how those data were used in the current paper.

(9) The examination of gamma functional connectivity in the 60 - 90 Hz range could be better motivated. Although some citations involving short-range connectivity in these frequencies are given (e.g., within the visual system), a more compelling argument for looking at this frequency range for longer-range connectivity may be required.

(10) The choice of source localization method (linearly constrained minimum variance) could be explained, particularly given that other methods (e.g. dynamic imaging of coherent sources) were specifically designed and might potentially be a better alternative for the types of analyses performed in the study.

(11) The mGC analysis needs to be more comprehensively detailed for the reader to be able to assess what is being reported and the strength of the evidence. Relatedly, first-level statistics (e.g., via estimation of the noise level) would make the mGC and DAI results more compelling.

(12) Considering the exploratory nature of the study, it is essential for other researchers to continue investigating and validating the results presented in the current manuscript. Thus, it is concerning that data and scripts are not fully and openly available. Data need not be in its raw state to be shared and useful, which circumvents the stated data privacy concerns.

---

## [Author Response]

**Reviewer #1 (Public Review):**
Abbasi et al. assess in this MEG study the directed connectivity of both cortical and subcortical regions during continuous speech production and perception. The authors observed bidirectional connectivity patterns between speech-related cortical areas as well as subcortical areas in production and perception. Interestingly, they found in speaking low-frequency connectivity from subcortical (the right cerebellum) to cortical (left superior temporal) areas, while connectivity from the cortical to subcortical areas was in the high frequencies. In listening a similar cortico-subcortical connectivity pattern was observed for the low frequencies, but the reversed connectivity in the higher frequencies was absent.The work by Abbasi and colleagues addresses a relevant, novel topic, namely understanding the brain dynamics between speaking and listening. This is important because traditionally production and perception of speech and language are investigated in a modality-specific manner. To have a more complete understanding of the neurobiology underlying these different speech behaviors, it is key to also understand their similarities and differences. Furthermore, to do so, the authors utilize state-of-the-art directed connectivity analyses on MEG measurements, providing a quite detailed profile of cortical and subcortical interactions for the production and perception of speech. Importantly, and perhaps most interesting in my opinion, is that the authors find evidence for frequency-specific directed connectivity, which is (partially) different between speaking and listening. This could suggest that both speech behaviors rely (to some extent) on similar cortico-cortical and cortico-subcortical networks, but different frequency-specific dynamics.These elements mentioned above (investigation of both production and perception, both cortico-cortical and cortico-subcortical connectivity is considered, and observing frequency-specific connectivity profiles within and between speech behaviors), make for important novel contributions to the field. Notwithstanding these strengths, I find that they are especially centered on methodology and functional anatomical description, but that precise theoretical contributions for neurobiological and cognitive models of speech are less transparent. This is in part because the study compares speech production and perception in general, but no psychophysical or psycholinguistic manipulations are considered. I also have some critical questions about the design which may pose some confounds in interpreting the data, especially with regard to comparing production and perception.(1) While the cortico-cortical and cortico-subcortical connectivity profiles highlighted in this study and the depth of the analyses are impressive, what these data mean for models of speech processing remains on the surface. This is in part due, I believe, to the fact that the authors have decided to explore speaking and listening in general, without targeting specific manipulations that help elucidate which aspects of speech processing are relevant for the particular connectivity profiles they have uncovered. For example, the frequency-specific directed connectivity is it driven by low-level psychophysical attributes of the speech or by more cognitive linguistic properties? Does it relate to the monitoring of speech, timing information, and updating of sensory predictions? Without manipulations trying to target one or several of these components, as some of the referenced work has done (e.g., Floegel et al., 2020; Stockert et al., 2021; Todorović et al., 2023), it is difficult to draw concrete conclusions as to which representations and/or processes of speech are reflected by the connectivity profiles. An additional disadvantage of not having manipulations within each speech behavior is that it makes the comparison between listening and speaking harder. That is, speaking and listening have marked input-output differences which likely will dominate any comparison between them. These physically driven differences (or similarities for that matter; see below) can be strongly reduced by instead exploring the same manipulations/variables between speaking and listening. If possible (if not to consider for future work), it may be interesting to score psychophysical (e.g., acoustic properties) or psycholinguistic (e.g., lexical frequency) information of the speech and see whether and how the frequency-specific connectivity profiles are affected by it.

We thank the reviewer for pointing this out. The current study is indeed part of a larger project investigating the role of the internal forward model in speech perception and production. In the original, more comprehensive study, we also included a masked condition where participants produced speech as usual, but their auditory perception was masked. This allowed us to examine how the internal forward model behaves when it doesn't receive the expected sensory consequences of generated speech. However, for the current study, we focused solely on data from the speaking and listening conditions due to its specific research question. We agree that further manipulations would be interesting. However, for this study our focus was on natural speech and we avoided other manipulations (beyond masked speech) so that we can have sufficiently long recording time for the main speaking and listening conditions.

(2) Recent studies comparing the production and perception of language may be relevant to the current study and add some theoretical weight since their data and interpretations for the comparisons between production and perception fit quite well with the observations in the current work. These studies highlight that language processes between production and perception, specifically lexical and phonetic processing (Fairs et al., 2021), and syntactic processing (Giglio et al., 2024), may rely on the same neural representations, but are differentiated in their (temporal) dynamics upon those shared representations. This is relevant because it dispenses with the classical notion in neurobiological models of language where production and perception rely on (partially) dissociable networks (e.g., Price, 2010). Rather those data suggest shared networks where different language behaviors are dissociated in their dynamics. The speech results in this study nicely fit and extend those studies and their theoretical implications.

We thank the reviewer for the suggestion and we will include these references and the points made by the reviewer in our revised manuscript.

(3) The authors align the frequency-selective connectivity between the right cerebellum and left temporal speech areas with recent studies demonstrating a role for the right cerebellum for the internal modelling in speech production and monitoring (e.g., Stockert et al., 2021; Todorović et al., 2023). This link is indeed interesting, but it does seem relevant to point out that at a more specific scale, it does not concern the exact same regions between those studies and the current study. That is, in the current study the frequency-specific connectivity with temporal regions concerns lobule VI in the right cerebellum, while in the referenced work it concerns Crus I/II. The distinction seems relevant since Crus I/II has been linked to the internal modelling of more cognitive behavior, while lobule VI seems more motor-related and/or contextual-related (e.g., D'Mello et al., 2020; Runnqvist et al., 2021; Runnqvist, 2023).

We thank the reviewer for their insightful comment. The reference was intended to provide evidence for the role of the cerebellum in internal modelling in speech. We do not claim that we have the spatial resolution with MEG to reliably spatially resolve specific parts of the cerebellum.

(4) On the methodological side, my main concern is that for the listening condition, the authors have chosen to play back the speech produced by the participants in the production condition. Both the fixed order as well as hearing one's own speech as listening condition may produce confounds in data interpretation, especially with regard to the comparison between speech production and perception. Could order effects impact the observed connectivity profiles, and how would this impact the comparison between speaking and listening? In particular, I am thinking of repetition effects present in the listening condition as well as prediction, which will be much more elevated for the listening condition than the speaking condition. The fact that it also concerns their own voice furthermore adds to the possible predictability confound (e.g., Heinks-Maldonado et al., 2005). In addition, listening to one's speech which just before has been articulated may, potentially strategically even, enhance inner speech and "mouthing" in the participants, hereby thus engaging the production mechanism. Similarly, during production, the participants already hear their own voice (which serves as input in the subsequent listening condition). Taken together, both similarities or differences between speaking and listening connectivity may have been due to or influenced by these order effects, and the fact that the different speech behaviors are to some extent present in both conditions.

This is a valid point raised by the reviewer. By listening to their own previously produced speech, our participants might have anticipated and predicted the sentences easier. However, during designing our experiment, we tried to lower the chance of this anticipation by several steps. First, participants were measured in separate sessions for speech production and perception tasks. There were always several days' intervals between performing these two conditions. Secondly, our questions were mainly about a common/general topic. Consequently, participants may not remember their answers completely.

Importantly, using the same stimulus material for speaking and listening guaranteed that there was no difference in the low-level features of the material for both conditions that could have affected the results of our statistical comparison.

Due to bone conduction, hearing one’s unaltered own speech from a recording may seem foreign and could lead to unwanted emotional reactions e.g. embarrassment, so participants were asked whether they heard their own voice in a recording already (e.g. from a self-recorded voice-message in WhatsApp) which most of them confirmed. Participants were also informed that they were going to hear themselves during the measurement to further reduce unwanted psychophysiological responses.

(5) The ability of the authors to analyze the spatiotemporal dynamics during continuous speech is a potentially important feat of this study, given that one of the reasons that speech production is much less investigated compared to perception concerns motor and movement artifacts due to articulation (e.g., Strijkers et al., 2010). Two questions did spring to mind when reading the authors' articulation artifact correction procedure: If I understood correctly, the approach comes from Abbasi et al. (2021) and is based on signal space projection (SSP) as used for eye movement corrections, which the authors successfully applied to speech production. However, in that study, it concerned the repeated production of three syllables, while here it concerns continuous speech of full words embedded in discourse. The articulation and muscular variance will be much higher in the current study compared to three syllables (or compared to eye movements which produce much more stable movement potentials compared to an entire discourse). Given this, I can imagine that corrections of the signal in the speaking condition were likely substantial and one may wonder (1) how much signal relevant to speech production behavior is lost?; (2) similar corrections are not necessary for perception, so how would this marked difference in signal processing affect the comparability between the modalities?

One of the results of our previous study (Abbasi et al., 2021) was that the artefact correction was not specific to individual syllables but generalised across syllables. Also, the repeated production of syllables was associated with substantial movements of the articulators mimicking those observed during naturalistic speaking. We therefore believe that the artefact rejection is effective during speaking. We also checked this by investigating speech related coherence in brain parcels in spatial proximity to the articulators.In our previous study we also show that the correction method retains neural activity to a very large degree. We are therefore confident that speaking and listening conditions can be compared and that the loss of true signals from correcting the speaking data will be minor.

References:Abbasi, O., Steingräber, N., & Gross, J. (2021). Correcting MEG artifacts caused by overt speech. Frontiers in Neuroscience, 15, 682419.D'Mello, A. M., Gabrieli, J. D., & Nee, D. E. (2020). Evidence for hierarchical cognitive control in the human cerebellum. Current Biology, 30(10), 1881-1892.Fairs, A., Michelas, A., Dufour, S., & Strijkers, K. (2021). The same ultra-rapid parallel brain dynamics underpin the production and perception of speech. Cerebral Cortex Communications, 2(3), tgab040.Floegel, M., Fuchs, S., & Kell, C. A. (2020). Differential contributions of the two cerebral hemispheres to temporal and spectral speech feedback control. Nature Communications, 11(1), 2839.Giglio, L., Ostarek, M., Sharoh, D., & Hagoort, P. (2024). Diverging neural dynamics for syntactic structure building in naturalistic speaking and listening. Proceedings of the National Academy of Sciences, 121(11), e2310766121.Heinks‐Maldonado, T. H., Mathalon, D. H., Gray, M., & Ford, J. M. (2005). Fine‐tuning of auditory cortex during speech production. Psychophysiology, 42(2), 180-190.Price, C. J. (2010). The anatomy of language: a review of 100 fMRI studies published in 2009. Annals of the new York Academy of Sciences, 1191(1), 62-88.Runnqvist, E., Chanoine, V., Strijkers, K., Pattamadilok, C., Bonnard, M., Nazarian, B., ... & Alario, F. X. (2021). Cerebellar and cortical correlates of internal and external speech error monitoring. Cerebral Cortex Communications, 2(2), tgab038.Runnqvist, E. (2023). Self-monitoring: The neurocognitive basis of error monitoring in language production. In Language production (pp. 168-190). Routledge.Stockert, A., Schwartze, M., Poeppel, D., Anwander, A., & Kotz, S. A. (2021). Temporo-cerebellar connectivity underlies timing constraints in audition. Elife, 10, e67303.Strijkers, K., Costa, A., & Thierry, G. (2010). Tracking lexical access in speech production: electrophysiological correlates of word frequency and cognate effects. Cerebral cortex, 20(4), 912-928.Todorović, S., Anton, J. L., Sein, J., Nazarian, B., Chanoine, V., Rauchbauer, B., ... & Runnqvist, E. (2023). Cortico-cerebellar monitoring of speech sequence production. Neurobiology of Language, 1-21.
**Reviewer #2 (Public Review):**
Summary:The authors re-analyse MEG data from a speech production and perception study and extend their previous Granger causality analysis to a larger number of cortical-cortical and in particular cortical-subcortical connections. Regions of interest were defined by means of a meta-analysis using Neurosynth.org and connectivity patterns were determined by calculating directed influence asymmetry indices from the Granger causality analysis results for each pair of brain regions. Abbasi et al. report feedforward signals communicated via fast rhythms and feedback signals via slow rhythms below 40 Hz, particularly during speaking. The authors highlight one of these connections between the right cerebellum lobule VI and auditory association area A5, where in addition the connection strength correlates negatively with the strength of speech tracking in the theta band during speaking (significant before multiple comparison correction). Results are interpreted within a framework of active inference by minimising prediction errors.While I find investigating the role of cortical-subcortical connections in speech production and perception interesting and relevant to the field, I am not yet convinced that the methods employed are fully suitable to this endeavour or that the results provide sufficient evidence to make the strong claim of dissociation of bottom-up and top-down information flow during speaking in distinct frequency bands.Strengths:The investigation of electrophysiological cortical-subcortical connections in speech production and perception is interesting and relevant to the field. The authors analyse a valuable dataset, where they spent a considerable amount of effort to correct for speech production-related artefacts. Overall, the manuscript is well-written and clearly structured.Weaknesses:The description of the multivariate Granger causality analysis did not allow me to fully grasp how the analysis was performed and I hence struggled to evaluate its appropriateness.Knowing that (1) filtered Granger causality is prone to false positives and (2) recent work demonstrates that significant Granger causality can simply arise from frequency-specific activity being present in the source but not the target area without functional relevance for communication (Schneider et al. 2021) raises doubts about the validity of the results, in particular with respect to their frequency specificity. These doubts are reinforced by what I perceive as an overemphasis on results that support the assumption of specific frequencies for feedforward and top-down connections, while findings not aligning with this hypothesis appear to be underreported. Furthermore, the authors report some main findings that I found difficult to reconcile with the data presented in the figures. Overall, I feel the conclusions with respect to frequency-specific bottom-up and top-down information flow need to be moderated and that some of the reported findings need to be checked and if necessary corrected.Major points(1) I think more details on the multivariate GC approach are needed. I found the reference to Schaum et al., 2021 not sufficient to understand what has been done in this paper. Some questions that remained for me are:(i) Does multivariate here refer to the use of the authors' three components per parcel or to the conditioning on the remaining twelve sources? I think the latter is implied when citing Schaum et al., but I'm not sure this is what was done here?If it was not: how can we account for spurious results based on indirect effects?

Yes, multivariate refers to the three components.

(ii) Did the authors check whether the GC of the course-target pairs was reliably above the bias level (as Schaum et. al. did for each condition separately)? If not, can they argue why they think that their results would still be valid? Does it make sense to compute DAIs on connections that were below the bias level? Should the data be re-analysed to take this concern into account?

We performed statistics on DAI and believe that this is a valid approach. We argue that random GC effects would not survive our cluster-corrected statistics.

(iii) You may consider citing the paper that introduced the non-parametric GC analysis (which Schaum et al. then went on to apply): Dhamala M, Rangarajan G, Ding M. Analyzing Information Flow in Brain Networks with Nonparametric Granger Causality. Neuroimage. 2008; 41(2):354-362. https://doi.org/10.1016/j.neuroimage.2008.02. 020

Thanks, we will add this reference in the revised version.

(2) GC has been discouraged for filtered data as it gives rise to false positives due to phase distortions and the ineffectiveness of filtering in the information-theoretic setting as reducing the power of a signal does not reduce the information contained in it (Florin et al., 2010; Barnett and Seth, 2011; Weber et al. 2017; Pinzuti et al., 2020 - who also suggest an approach that would circumvent those filter-related issues). With this in mind, I am wondering whether the strong frequency-specific claims in this work still hold.

This must be a misunderstanding. We are aware of the problem with GC on filtered data. But GC was here computed on broadband data and not in individual frequency bands.

(3) I found it difficult to reconcile some statements in the manuscript with the data presented in the figures:(i) Most notably, the considerable number of feedforward connections from A5 and STS that project to areas further up the hierarchy at slower rhythms (e.g. L-A5 to R-PEF, R-Crus2, L CB6 L-Tha, L-FOP and L-STS to R-PEF, L-FOP, L-TOPJ or R-A5 as well as R-STS both to R-Crus2, L-CB6, L-Th) contradict the authors' main message that 'feedback signals were communicated via slow rhythms below 40 Hz, whereas feedforward signals were communicated via faster rhythms'. I struggled to recognise a principled approach that determined which connections were highlighted and reported and which ones were not.(ii) "Our analysis also revealed robust connectivity between the right cerebellum and the left parietal cortex, evident in both speaking and listening conditions, with stronger connectivity observed during speaking. Notably, Figure 4 depicts a prominent frequency peak in the alpha band, illustrating the specific frequency range through which information flows from the cerebellum to the parietal areas." There are two peaks discernible in Figure 4, one notably lower than the alpha band (rather theta or even delta), the other at around 30 Hz. Nevertheless, the authors report and discuss a peak in the alpha band.(iii) In the abstract: "Notably, high-frequency connectivity was absent during the listening condition." and p.9 "In contrast with what we reported for the speaking condition, during listening, there is only a significant connectivity in low frequency to the left temporal area but not a reverse connection in the high frequencies."While Fig. 4 shows significant connectivity from R-CB6 to A5 in the gamma frequency range for the speaking, but not for the listening condition, interpreting comparisons between two effects without directly comparing them is a common statistical mistake (Makin and Orban de Xivry). The spectrally-resolved connectivity in the two conditions actually look remarkably similar and I would thus refrain from highlighting this statement and indicate clearly that there were no significant differences between the two conditions.(iv) "This result indicates that in low frequencies, the sensory-motor area and cerebellum predominantly transmit information, while in higher frequencies, they are more involved in receiving it."I don't think that this statement holds in its generality: L-CB6 and R-3b both show strong output at high frequencies, particularly in the speaking condition. While they seem to transmit information mainly to areas outside A5 and STS these effects are strong and should be discussed.

We appreciate the reviewer's thoughtful comments. We acknowledge that not all connectivity patterns strictly adhere to the initial observation regarding feedback and feedforward communication. It's true that our primary focus was on interactions between brain regions known to be crucial for speech prediction, including auditory, somatosensory, and cerebellar areas. However, we also presented connectivity patterns across other regions to provide a more comprehensive picture of the speech network. We believe this broader perspective can be valuable for future research directions.

Regarding the reviewer's observation about the alpha band peak in Figure 4, we agree that a closer examination reveals the connectivity from right cerebellum to the left parietal is in a wider low frequency range. We will refrain from solely emphasizing the alpha band and acknowledge the potential contribution of lower frequencies to cerebellar-parietal communication.

We also appreciate the reviewer highlighting the need for a more nuanced interpretation of the listening condition connectivity compared to the speaking condition. The reviewer is correct in pointing out that while Figure 4 suggests a high-frequency connectivity from L-A5 to R-CB only in the speaking condition, a direct statistical comparison between conditions might not reveal a significant difference. We will revise the manuscript to clarify this point.

Finally, a closer examination of Figure 3 revealed that the light purple and dark green edges in the speaking condition for R-CB6 and L-3b suggest outgoing connections at low frequencies, while other colored edges indicate information reception at high frequencies. We acknowledge that exceptions to this directional pattern might exist and warrant further investigation in future studies.

(4) "However, definitive conclusions should be drawn with caution given recent studies raising concerns about the notion that top-down and bottom-up signals can only be transmitted via separate frequency channels (Ferro et al., 2021; Schneider et al., 2021; Vinck et al., 2023)."I appreciate this note of caution and think it would be useful if it were spelled out to the reader why this is the case so that they would be better able to grasp the main concerns here. For example, Schneider et al. make a strong point that we expect to find Granger-causality with a peak in a specific frequency band for areas that are anatomically connected when the sending area shows stronger activity in that band than the receiving one, simply because of the coherence of a signal with its own linear projection onto the other area. The direction of a Granger causal connection would in that case only indicate that one area shows stronger activity than the other in the given frequency band. I am wondering to what degree the reported connectivity pattern can be traced back to regional differences in frequency-specific source strength or to differences in source strength across the two conditions.

This is indeed an important point. That is why we are discussing our results with great caution and specifically point the reader to the relevant literature. We are indeed thinking about a future study where we investigate this connectivity using other connectivity metrics and a detailed consideration of power.

**Reviewer #3 (Public Review):**
In the current paper, Abbasi et al. aimed to characterize and compare the patterns of functional connectivity across frequency bands (1 Hz - 90 Hz) between regions of a speech network derived from an online meta-analysis tool (Neurosynth.org) during speech production and perception. The authors present evidence for complex neural dynamics from which they highlight directional connectivity from the right cerebellum to left superior temporal areas in lower frequency bands (up to beta) and between the same regions in the opposite direction in the (lower) high gamma range (60-90 Hz). Abbasi et al. interpret their findings within the predictive coding framework, with the cerebellum and other "higher-order" (motor) regions transmitting top-down sensory predictions to "lower-order" (sensory) regions in the lower frequencies and prediction errors flowing in the opposite direction (i.e., bottom-up) from those sensory regions in the gamma band. They also report a negative correlation between the strength of this top-down functional connectivity and the alignment of superior temporal regions to the syllable rate of one's speech.Strengths:(1) The comprehensive characterization of functional connectivity during speaking and listening to speech may be valuable as a first step toward understanding the neural dynamics involved.(2) The inclusion of subcortical regions and connectivity profiles up to 90Hz using MEG is interesting and relatively novel.(3) The analysis pipeline is generally adequate for the exploratory nature of the work.Weaknesses:(1) The work is framed as a test of the predictive coding theory as it applies to speech production and perception, but the methodological approach is not suited to this endeavor.

We agree that we cannot provide definite evidence for predictive coding in speech production and perception and we believe that we do not make that claim in the manuscript. However, our results are largely consistent with what can be expected based on predictive coding theory.

(2) Because of their theoretical framework, the authors readily attribute roles or hierarchy to brain regions (e.g., higher- vs lower-order) and cognitive functions to observed connectivity patterns (e.g., feedforward vs feedback, predictions vs prediction errors) that cannot be determined from the data. Thus, many of the authors' claims are unsupported.

We will revise the manuscript to more clearly differentiate our results (e.g. directed Granger-Causality from A to B) from their interpretation (potentially indicating feedforward or feedback signals).

(3) The authors' theoretical stance seems to influence the presentation of the results, which may inadvertently misrepresent the (otherwise perfectly valid; cf. Abbasi et al., 2023) exploratory nature of the study. Thus, results about specific regions are often highlighted in figures (e.g., Figure 2 top row) and text without clear reasons.

Our connectograms reveal a multitude of results that we hope is interesting to the community. At the same time the wealth of findings poses a problem for describing them. We did not see a better way then to highlight specific connections of interest.

(4) Some of the key findings (e.g., connectivity in opposite directions in distinct frequency bands) feature in a previous publication and are, therefore, interesting but not novel.

We actually see this as a strength of the current manuscript. The computation of connectivity is here extended to a much larger sample of brain areas. It is reassuring to see that the previously reported results generalise to other brain areas.

(5) The quantitative comparison between speech production and perception is interesting but insufficiently motivated.

We thank the reviewer for this comment. We have addressed that in detail in response to the point (1&4) of reviewer 1.

(6) Details about the Neurosynth meta-analysis and subsequent selection of brain regions for the functional connectivity analyses are incomplete. Moreover, the use of the term 'Speech' in Neurosynth seems inappropriate (i.e., includes irrelevant works, yielding questionable results). The approach of using separate meta-analyses for 'Speech production' and 'Speech perception' taken by Abbasi et al. (2023) seems more principled. This approach would result, for example, in the inclusion of brain areas such as M1 and the BG that are relevant for speech production.

We agree that there are inherent limitations in automated meta-analysis tools such as Neurosynth. Papers are used in the meta-analysis that might not be directly relevant. However, Neurosynth has proven its usefulness over many years and has been used in many studies. We also agree that our selection of brain areas is not complete. But Granger Causality analysis of every pair of ROIs leads to complex results and we had to limit our selection of areas.

(7) The results involving subcortical regions are central to the paper, but no steps are taken to address the challenges involved in the analysis of subcortical activity using MEG. Additional methodological detail and analyses would be required to make these results more compelling. For example, it would be important to know what the coverage of the MEG system is, what head model was used for the source localization of cerebellar activity, and if specific preprocessing or additional analyses were performed to ensure that the localized subcortical activity (in particular) is valid.

There is a large body of evidence demonstrating that MEG can record signals from deep brain areas such as thalamus and cerebellum including Attal & Schwarz 2013, Andersen et al, Neuroimage 2020; Piastra et al., 2020; Schnitzler et al., 2009. These and other studies provide evidence that state-of-the-art recording (with multichannel SQUID systems) and analysis is sufficient to allow reconstruction of subcortical areas. However, spatial resolution is clearly reduced for these deep areas. We will add a statement in the revised manuscript to acknowledge this limitation.

(8) The results and methods are often detailed with important omissions (a speech-brain coupling analysis section is missing) and imprecisions (e.g., re: Figure 5; the Connectivity Analysis section is copy-pasted from their previous work), which makes it difficult to understand what is being examined and how. (It is also not good practice to refer the reader to previous publications for basic methodological details, for example, about the experimental paradigm and key analyses.) Conversely, some methodological details are given, e.g., the acquisition of EMG data, without further explanation of how those data were used in the current paper.

We will revise the relevant sections of the manuscript.

(9) The examination of gamma functional connectivity in the 60 - 90 Hz range could be better motivated. Although some citations involving short-range connectivity in these frequencies are given (e.g., within the visual system), a more compelling argument for looking at this frequency range for longer-range connectivity may be required.

Given previous evidence of connectivity in the gamma band we think that it would be a weakness to exclude this frequency band from analysis.

(10) The choice of source localization method (linearly constrained minimum variance) could be explained, particularly given that other methods (e.g. dynamic imaging of coherent sources) were specifically designed and might potentially be a better alternative for the types of analyses performed in the study.

Both LCMV and DICS are beamforming methods. We used LCMV because we wanted used Granger Causality which requires broadband signals. DICS would only provide frequency-specific band-limited signals.

(11) The mGC analysis needs to be more comprehensively detailed for the reader to be able to assess what is being reported and the strength of the evidence. Relatedly, first-level statistics (e.g., via estimation of the noise level) would make the mGC and DAI results more compelling.

We perform group-level cluster-based statistics on mGC while correcting for multiple comparisons across frequency bands and brain parcels and report only significant results. This is an established approach that is routinely used in this type of studies.

(12) Considering the exploratory nature of the study, it is essential for other researchers to continue investigating and validating the results presented in the current manuscript. Thus, it is concerning that data and scripts are not fully and openly available. Data need not be in its raw state to be shared and useful, which circumvents the stated data privacy concerns.

We acknowledge the reviewer's concern regarding the full availability of the dataset. Due to privacy limitations on the collected data, we are unable to share it publicly at this time. However, to promote transparency and enable further exploration, we have provided the script used for data analysis and an example dataset. This example dataset should provide a clear understanding of the data structure and variables used in the analysis. Additionally, we are happy to share the complete dataset upon request from research teams interested in performing in-depth secondary analyses.